# Oral Candidiasis: A Disease of Opportunity

**DOI:** 10.3390/jof6010015

**Published:** 2020-01-16

**Authors:** Taissa Vila, Ahmed S. Sultan, Daniel Montelongo-Jauregui, Mary Ann Jabra-Rizk

**Affiliations:** 1Department of Oncology and Diagnostic Sciences, School of Dentistry, University of Maryland, Baltimore, MD 21201, USA; tvila@umaryland.edu (T.V.); asultan@umaryland.edu (A.S.S.); dmontelongo@umaryland.edu (D.M.-J.); 2Department of Microbiology and Immunology, School of Medicine, University of Maryland, Baltimore, MD 21201, USA

**Keywords:** oral candidiasis, *Candida albicans*, immune response, fungal–bacterial interactions

## Abstract

Oral candidiasis, commonly referred to as “thrush,” is an opportunistic fungal infection that commonly affects the oral mucosa. The main causative agent, *Candida albicans*, is a highly versatile commensal organism that is well adapted to its human host; however, changes in the host microenvironment can promote the transition from one of commensalism to pathogen. This transition is heavily reliant on an impressive repertoire of virulence factors, most notably cell surface adhesins, proteolytic enzymes, morphologic switching, and the development of drug resistance. In the oral cavity, the co-adhesion of *C. albicans* with bacteria is crucial for its persistence, and a wide range of synergistic interactions with various oral species were described to enhance colonization in the host. As a frequent colonizer of the oral mucosa, the host immune response in the oral cavity is oriented toward a more tolerogenic state and, therefore, local innate immune defenses play a central role in maintaining *Candida* in its commensal state. Specifically, in addition to preventing *Candida* adherence to epithelial cells, saliva is enriched with anti-candidal peptides, considered to be part of the host innate immunity. The T helper 17 (Th17)-type adaptive immune response is mainly involved in mucosal host defenses, controlling initial growth of *Candida* and inhibiting subsequent tissue invasion. Animal models, most notably the mouse model of oropharyngeal candidiasis and the rat model of denture stomatitis, are instrumental in our understanding of *Candida* virulence factors and the factors leading to host susceptibility to infections. Given the continuing rise in development of resistance to the limited number of traditional antifungal agents, novel therapeutic strategies are directed toward identifying bioactive compounds that target pathogenic mechanisms to prevent *C. albicans* transition from harmless commensal to pathogen.

## 1. Introduction

### 1.1. *Candida* and Candidiasis Etymology and Historical Perspective

Oral candidiasis (OC), commonly referred to as “thrush” encompasses infections of the tongue and other oral mucosal sites and is characterized by fungal overgrowth and invasion of superficial tissues [1,2,3]. The colloquial term “thrush” refers to the resemblance of the white flecks present in some forms of candidiasis with the breast of the bird of the same name. The etymology of oral thrush dates back to the time of Hippocrates (around 400 Before Christ (BC)) who, in his book “*Of the Epidemics*,” described OC as “*mouths affected with aphthous ulcerations*” [4]. The early descriptions of the disease predated the concept of “contagion” and, therefore, as recently as the early 1900s, it was thought that the disease was of host origin.

Approximately 200 years were required before the etiological agent of thrush was correctly identified as a fungal pathogen. In 1771, Rosen von Rosenstein defined an invasive form of thrush; however, in 1839, Langenbeck was credited with first documenting a fungus associated with thrush in a patient with typhoid fever [5,6]. In 1846, Berg presented observations that thrush was caused by a fungus, which was classified in 1847 by the French mycologist, Charles Philippe Robin as *Oidium albicans*, the first use of albicans which means “to whiten” [6,7]. In 1923, Berkhout reclassified the fungus under the current genus *Candida*, a name derived from the Latin word *toga candida*, referring to the white toga (robe) worn by Roman senators of the ancient Roman republic, a probable reference to the whitish colonies on agar or white lesions [6,7,8]. However, it was not until 1954 that the binomial *Candida albicans* was formally endorsed. In the 1980s, there was a clear surge of interest in oral candidal infections largely due to the increased incidence of OC because of the escalation in the acquired immune deficiency syndrome (AIDS) epidemic, and, to date, OC remains the most common oral opportunistic infection in human immunodeficiency virus (HIV)-positive individuals and in individuals with weakened immune systems [9,10,11,12,13]. In fact, the opportunistic nature of the infection was first highlighted by Hippocrates, who referred to this malady as “*a disease of the diseased*” [14].

### 1.2. *Candida albicans*: An Opportunistic Pathogen

*C. albicans* is by far the main causative agent of OC accounting for up to 95% of cases. Although considered a pathogen, *C. albicans* is a ubiquitous commensal organism that commonly colonizes the oral mucosa and is readily isolated from the oral cavities of healthy individuals. In fact, up to 80% of the general population are asymptomatic carriers, and simple carriage does not predictably lead to infection [15,16,17,18,19]. Similar to the oral cavity, *C. albicans* asymptomatically colonizes the gastrointestinal tract and reproductive tract of healthy individuals where its proliferation at these various sites is controlled by the host immune system, and other members of the microbiota [20,21]. Uniquely, *C. albicans* is a highly versatile commensal organism that is well adapted to its human host, and any changes in the host microenvironment that favor its proliferation provide this pathogen with the opportunity to invade virtually any site. This can manifest with superficial mucosal infections to invasive disseminated disease with involvement of multiple organs [10,14,15,22,23,24,25,26]. Notwithstanding, however, is the impressive repertoire of virulence factors that *C. albicans* possesses, enabling it to rapidly transition to a pathogen, the most notable of which are listed in Table 1 [27,28].

First and foremost, in order for *Candida* to cause infection, it has to be retained within the mouth. However, removal of loosely attached *Candida* cells from mucosal surfaces via the effects of salivary flow and swallowing is an important factor in host defense against *Candida* overgrowth [14]. Therefore, the ability to circumvent these removal mechanisms can be regarded as a key virulence attribute. Although, during its commensal yeast state, *C. albicans* reversibly adheres to oral epithelial cells through electrostatic interactions, attachment to oral epithelial surfaces is mediated by cell-wall receptors such as the agglutinin-like sequence (ALS) family of glycoproteins [15,16,29,30,31,32]. Most notable among the members of the family is the hyphal-specific adhesin Als3p, which was also shown to act as a receptor for bacterial adherence to *C. albicans* hyphae [33,34]. Similarly, the hyphal wall protein (Hwp1) is another major adhesin, and deletion of either *ALS3* or *HWP1* genes was shown to result in attenuated virulence [35,36].

Once attached to host surfaces, *C. albicans* can switch morphology to the invasive filamentous form which facilitates epithelial penetration [14,37]. In fact, core to *C. albicans* pathogenesis is its ability to undergo morphologic switching between yeast and hyphal forms [38,39,40,41,42,43]. Yeast-to-hypha transition is triggered in response to a variety of host environmental stimuli that activate multiple regulatory signaling pathways, eventually leading to the expression of master activators of hyphal formation [13,44]. The distinct morphological states of *C. albicans* dictate phases of colonization, growth, and dissemination, where the yeast form is associated with both initial attachment and dissemination, while the hyphal form enables *C. albicans* to invade host tissue [23,27,42,45]. In fact, hypha formation is associated with the expression of hypha-associated virulence factors that aid in adhesion to and invasion into host cells. One important property of hyphal cells is their ability of directional growth in response to contact with a surface (thigmotropism), allowing the fungus to specifically invade intercellular junctions [27,37]. In addition to active penetration which is a fungal-driven process, another complementary mechanism utilized by *C. albicans* for host cell invasion is endocytosis, a passive fungal-induced but host cell-driven process whereby lytic enzymes and invasins expressed on hyphae bind to and degrade E-cadherin and other inter-epithelial cell junctional proteins, enabling the organism to penetrate between epithelial cells [27,31,35].

Aside from the physical effect of filamentous growth, destruction of host tissue by *C. albicans* is augmented by extracellular hydrolytic enzymes released by the fungus into the local environment. Most notable of the extracellularly secreted enzymes frequently implicated in the virulence of *C. albicans* are secreted aspartyl proteinases (SAPs) and secreted phospholipases (PL), which are involved in host tissue invasion and nutrient acquisition [23,35,46,47]. Importantly, in addition to digesting and destroying cell membranes, SAPs also allow *C. albicans* to evade host defenses by degrading molecules of the host immune system, including antibodies and antimicrobial peptides [14,46]. Interestingly, recently it was discovered that hyphae-induced epithelial damage was mainly mediated through the secretion of a cytolytic peptide toxin called candidalysin, encoded by the hyphal-specific gene *ECE1*. The importance of this newly identified virulence factor was clearly established when *C. albicans* mutants were found to be incapable of inducing tissue damage, and were highly attenuated in a mouse model of oropharyngeal candidiasis [35,48,49].

The major biological feature of *C. albicans* with significant clinical implications resides in its ability to form biofilms [38,41,50]. In fact, the majority of *C. albicans* infections are associated with formation of biofilms on a variety of surfaces, and the transition of *C. albicans* from budding yeast to a filamentous hyphal is central to its ability to form pathogenic biofilms [38,40,41,42,51,52,53]. Biofilms are structured communities of surface-associated microbial populations embedded in an extracellular matrix which are described to have a multifaceted role [42,54,55,56,57]. The *C. albicans* biofilm matrix is largely composed of the polysaccharides β-1,3-glucan, β-1,6-glucan, and mannans which form the mannan–glucan complex (MGCx) [58,59,60,61]. In the oral cavity, hyphae formation and adherence to oral epithelial cells and other abiotic surfaces such as dentures promotes the development of monomicrobial and polymicrobial biofilms [62,63]. Once a biofilm is established, the expression of *Candida* virulence factors increases, and susceptibility to antimicrobials and phagocytosis decreases drastically [23,41,64]. Importantly, in addition to *Candida* pathogenic factors and interactions with the host immune system, it is now acknowledged that the bacterial component of the oral microbiome plays an important role in the development and exacerbation of OC [14,65,66].

## 2. Clinical Manifestations of Oral Candidiasis

As the primary reservoir for oral *Candida* carriage, the tongue dorsum is the initiating point of infection for the majority of the clinical forms of oral candidiasis (OC) [67]. This includes oropharyngeal candidiasis (OPC) (Figure 1A), characterized by invasion of the epithelial cell lining of the oropharynx, which often occurs as an extension of OC. There are multiple clinical presentations and several classification systems for OC; however, the most simplistic classification encompasses oral manifestations that can generally be classified into three main broad categories, namely, (**1**) acute manifestations, (**2**) chronic manifestations, and (**3**) chronic mucocutaneous candidiasis syndromes. It is important to note that several clinical forms can occur in the oral cavity and in multiple oral sites at one time [67]. Additionally, although other non-albicans *Candida* species can cause OC, the oral manifestations are identical, irrespective of the causative species.

### 2.1. Acute Manifestations of Oral Candidiasis

#### 2.1.1. Acute Pseudomembranous Candidiasis

Acute pseudomembranous candidiasis, often referred to as “thrush”, usually presents as multifocal curdy yellow-white plaques throughout the oral mucosa (Figure 1A,B). A diagnostic feature of this infection is that these plaques, consisting of desquamated epithelial and immune cells together with yeast and hyphae, can be removed by gentle scraping, leaving behind an underlying red erosive base [1,3,14]. The diagnosis of pseudomembranous candidiasis is essentially a clinical diagnosis based on the presence of distinctive clinical features. Alternatively, a swab from the white patches can be sent for microscopic identification of *Candida* or for culture to identity the *Candida* species present [68,69]. Although the pseudomembranous candidiasis form is common in neonates and the vast majority of cases are due to the use of inhaled steroids, there is a direct relationship with immunodeficiency. In fact, pseudomembranous candidiasis is considered the main opportunistic infection in patients with AIDS and cancer, and in patients receiving immunosuppressive therapies. In the case of AIDS, chronic and recurrent infection is frequent, which can subsequently progress to esophageal candidiasis leading to difficulties in swallowing and nutrition.

#### 2.1.2. Acute Erythematous Candidiasis

Acute erythematous candidiasis is historically referred to as “antibiotic sore mouth” as it frequently occurs as a consequence of the reduction in levels of the bacterial oral microflora following broad-spectrum antibiotics which facilitates overgrowth of *Candida.* Cessation of antibiotic therapy restores the normal homeostatic balance of the microbial community, which subsequently resolves the infection without the need for therapeutic intervention [14]. This form of OC presents as painful reddened lesions throughout the oral cavity; lesions can either arise de novo or subsequent to shedding of the pseudomembrane from of acute pseudomembranous candidiasis [3,14,70].

### 2.2. Chronic Manifestations of Oral Candidiasis

#### 2.2.1. Chronic Erythematous Atrophic Candidiasis

Chronic erythematous atrophic candidiasis presents similarly to the acute form and usually occurs as an extension of it. This form is also often encountered in HIV^+^ individuals. The most prevalent form of chronic erythematous/atrophic candidiasis is *Candida*-associated denture stomatitis (DS), which most commonly presents as erythema of the denture-bearing palatal mucosa (Figure 1C). DS is seen in up to 75% of denture wearers and, often, there are no clinical symptoms [23,71,72,73]. Inadequate denture hygiene, ill-fitting dentures, or continuous wearing of dentures (especially nocturnal use) are the main host predisposing factors to DS [74,75]. Under these conditions, coupled with the limited flow of saliva at this location, the stagnant area beneath the denture provides an ideal environment for the growth of *Candida*. Frictional irritation by ill-fitting dentures can damage the mucosal barrier, allowing infiltration of colonizing *Candida* into the tissue causing infection [15,23]. Additionally, the abiotic acrylic material acts as a chronic reservoir allowing continuous seeding of *Candida* onto the palatal tissue; this in turn elicits a robust local inflammatory response that clinically manifests as tissue erythema and hyperplasia [14,23]. Given the propensity of *Candida* to adhere to and colonize the denture, this condition is considered a classic *Candida* biofilm-associated infection. In fact, *C. albicans* is recovered more frequently from the denture surface than from the associated palatal mucosa and, therefore, clinical management is primarily focused on eradication of the biofilm formed on the denture to prevent re-colonization and relapse [73,76].

#### 2.2.2. Angular Cheilitis

Angular cheilitis, as the term implies, affects the angles or commissures of the mouth and presents with erythema, maceration, fissuring, crusting, or a combination thereof (Figure 1D). The presentation may be unilateral but is more often bilateral. Angular cheilitis is commonly associated with DS or another pre-existing primary form of OC where the elevated numbers of *Candida* in the oral cavity result in direct spread and auto-inoculation of the angles of the mouth [3,14]. Furthermore, it is not uncommon for these lesions to be co-infected with *Staphylococcus aureus* and, therefore, the exact role that *Candida* itself plays in the infection is difficult to ascertain. One important predisposing factor is the reduced vertical occlusal dimension in elderly edentulous patients, predisposing individuals to exuberant redundant folds and maceration. Importantly, angular cheilitis can be secondary to hematinic deficiencies warranting further investigation with blood tests.

#### 2.2.3. Cheilocandidiasis

Cheilocandidiasis is a recently recognized form of chronic candidiasis that features crusting and ulcerations of the lips [70]. *Candida* thrives in moist environments and, therefore, cheilocandidiasis occurs as a consequence of continuous applications of petrolatum-based products, chronic lip-licking, or thumb-sucking. These and other factors that promote moist environments can cause pre-existing angular cheilitis to extend into the perioral skin [77].

#### 2.2.4. Chronic Hyperplastic Candidiasis

Chronic hyperplastic candidiasis, also referred to as candidal leukoplakia, usually arises on the anterior buccal mucosa proximal to the anterior commissures (retrocommissural area), but may also occur on the lateral tongue which is the second most common site of occurrence [78]. Patients present with well-demarcated leukoplakias or raised fissured white plaques that cannot be removed by gentle scraping. The highest prevalence of this rare form of OC is in middle-aged male smokers. An important consideration of chronic hyperplastic candidiasis is its association with an increased risk of malignant transformation (up to 10%) to oral squamous cell carcinoma, although the exact mechanism is currently unknown [79].

#### 2.2.5. Median Rhomboid Glossitis

Median rhomboid glossitis, also referred to as atrophic glossitis or central papillary atrophy, presents as a central elliptical or rhomboid area of atrophy and erythema of the midline posterior tongue dorsum, anterior to the circumvallate papillae [3,78]. This lesion was historically attributed to a developmental origin; however, this is unlikely as pediatric cases are seldom encountered. This condition is often associated with frequent use of steroid inhalers or tobacco smoking [14].

### 2.3. Chronic Mucocutaneous Candidiasis Syndromes

Chronic mucocutaneous candidiasis syndromes represent a group of several very rare heterogeneous immunologic disorders characterized by underlying immune deficiencies. Clinically, affected patients suffer from chronic and sometimes life-long persistent or recurrent mucocutaneous candidiasis involving the skin, nails, and genital mucosa; however, greater than 90% of patients present with oral involvement [80,81]. It is thought that the severity of the clinical manifestation correlates with the severity of the underlying immune defect. Many types of chronic mucocutaneous candidiasis syndromes exist that include the sporadic form, forms secondary to immunosuppressive therapies, diabetes, T-cell deficiency or HIV infection, inherited familial genetic forms, and autoimmune polyendocrinopathy candidiasis ectodermal dystrophy (APECED) [80,81]. These patients are often refractory to standard antifungal therapies and have an increased susceptibility to developing oral squamous cell carcinoma [70,81].

## 3. Predisposing Factors to Oral Candidiasis

The transition of *C. albicans* from a harmless commensal state to a pathogenic state is heavily reliant on many predisposing factors. The continued rise in the incidence of candidiasis is reflective of the increased use of broad-spectrum antibiotics, immunosuppressive agents, indwelling medical devices and catheters, and the increase in solid organ and hematopoietic cell transplantations [15]. The non-exhaustive list in Table 2 details the various predisposing factors and/or conditions that are involved in the development of many of the clinical forms of OC [15,23,82,83].

### 3.1. Local Factors

#### 3.1.1. Salivary Hypofunction

Saliva is enriched with antimicrobial proteins that aid in limiting *C. albicans* attachment to the oral epithelia, and this biofluid is largely responsible for the maintenance of *C. albicans* in its commensal state [84]. Therefore, quantitative and qualitative reductions in saliva are common factors implicated in the development of OC [85]. The incidence of salivary hypofunction is increasing due to the aging population and the increase in polypharmacy. Additionally, weakened immune states (e.g., HIV) and other iatrogenic therapies such as chemotherapy and head-and-neck radiation therapy result in profound insult to the salivary glands and contribute to the development of OC [86,87].

#### 3.1.2. Denture Wearing

Prolonged denture wearing, poor denture hygiene, and mucosal trauma are important local factors that contribute to OC development, as a breach in the oral epithelium creates a portal of entry for *Candida*. An important contributing factor to the development of DS is the favorable environment for *Candida* growth that is created beneath the dentures. The micro-environment of the denture-bearing palatal mucosa is of low oxygen, largely devoid of saliva, and is of low acidic pH, which promotes SAP activity [46]. DS affects both immunocompetent and immunocompromised patients but is invariably more common in elderly and immunocompromised individuals in which recurrent episodes are frequent. In fact, it was reported that at least 40% of elderly denture wearers do not adequately disinfect or remove their dentures at night, and life-threatening pneumonia events are twice as likely to occur in these patients [88]. Newton’s classification, introduced in 1962, is currently the most widely used clinical classification system for DS. The classification system is composed of three main clinical types: (**I**) pin-point erythema of the palatal mucosa, (**II**) diffuse erythema of the palatal mucosa, and (**III**) granular-type inflammatory papillary hyperplasia [89]. Studies demonstrated that patients who harbor mixed *Candida* species biofilms have an approximate five-fold increased risk of more severe disease (Newton’s type III DS), whereas patients solely colonized by *C. albicans* are three times as likely to manifest with less severe disease (Newton’s type I DS) [90]. Importantly, the type of denture material strongly influences biofilm development with acrylic dentures incurring a five-fold increase risk of DS as compared to metallic dentures [91]. Clinical findings similar to DS were also reported in patients wearing obturators or patients wearing orthodontic appliances [78].

#### 3.1.3. Topical Corticosteroid Therapy

Topical corticosteroid therapy is the mainstay for the management of chronic inflammatory oral mucosal diseases. It is important to note that severe oral mucosal disease, especially with extension to extra-oral sites, usually warrants systemic corticosteroid therapy. Patients may be managed with topical and systemic corticosteroids for long periods of time, often necessitating anti-fungal prophylaxis. Incorrect steroid inhaler use may also predispose to the development of OC as a consequence of suppressing cellular immunity and phagocytosis; however, the local mucosal immunity reverts to normal on discontinuation of the inhaled steroids. Local alterations in the oral environment arise from the immunosuppressive effects of these therapies and consequently give rise to secondary OC [92].

#### 3.1.4. Smoking

Tobacco cigarette users are known to have significantly higher oral candidal carriage levels and, therefore, are at an increased risk of developing OC [93]. However, newer non-conventional tobacco substitutes such as electronic nicotine delivery systems (ENDS) and the role they play as an etiologic factor in the development of OC are currently unknown [94]. Moreover, studies are needed to characterize the oral mycobiome and to determine if ENDS users are colonized with potentially carcinogenic *Candida* strains. Interestingly, a recent in vitro study indicated that ENDS can induce the expression of *C. albicans* virulence factors such as *SAP*2, *SAP*3, and *SAP*9 genes [95]. The exact mechanism via which conventional tobacco cigarette smoking predisposes to OC development is yet to be definitively established, but a plausible theory suggests that the decreased salivary flow rate in cigarette smokers and, consequently, the lowered pH may result in an acidic environment that is likely to favor *Candida* colonization and growth [96]. Additionally, it was suggested that cigarette smoking may cause a decrease in salivary immunoglobulin A (IgA) and a depression of neutrophil function, encouraging oral colonization of *Candida* [80]. Intriguingly, it was further theorized that cigarette smoking can provide nutrition for *Candida* to produce carcinogens [80,97].

### 3.2. Systemic Factors

#### 3.2.1. Age-Related Immunosenescence

Elderly patients were shown to have significantly lower activity levels of protective salivary innate defenses [98,99]. Moreover, infants at the other extreme of age are at increased risk for the development OC [70].

#### 3.2.2. Broad-Spectrum Antibiotics

Broad-spectrum antibiotics are responsible for the overwhelming majority of acute OC cases [100]. Dysbiosis by bacterial depletion due to the use of broad-spectrum antibiotics can alter the local oral flora, creating a favorable environment for *Candida* to proliferate.

#### 3.2.3. HIV Infection and AIDS

It is well established that HIV^+^ patients harbor increased levels of *Candida* colonizing the oral cavity and are significantly predisposed to OC [101]. Specifically, *C. dubliniensis* was recognized to have a strong proclivity for causing OC in HIV^+^ patients [101] with the corollary that cluster of differentiation 4 (CD4) T-cell levels are directly proportional to the severity of OC in this patient population [102]. Furthermore, HIV^+^ patients have significantly lower protective levels of antimicrobial peptides, namely, histatin-5 (Hst-5); thus, these patients are reported to have increased rates of OC compared to matched healthy controls [87]. Interestingly, linear gingival erythema was identified as a specific *Candida*-associated clinical finding in HIV^+^ patients [70]; linear gingival erythema clinically presents as a localized or generalized well-demarcated linear band of erythema along the free gingival margins [80]. Both antifungal therapy and adequate oral hygiene practices are required to eliminate this condition [78].

#### 3.2.4. Systemic Immunocompromise

Aside from HIV disease, any systemic disease that results in systemic immunocompromise, whether the underlying etiology is developmental, iatrogenic, immune-mediated, autoimmune, endocrine, or associated with a malignancy state, may give rise to OC. A brief non-exhaustive list includes systemic immunocompromise as a result of thymic aplasia, chronic mucocutaneous candidiasis syndromes, chemoradiation, cytotoxic therapies, immunomodulating agents, graft-versus-host disease, Sjogren’s syndrome, agranulocytosis, leukemia, diabetes mellitus, Addison’s disease, and hypothyroidism [70].

#### 3.2.5. Nutritional Deficiencies

Malnutrition, malabsorption, and eating disorder states are reported to predispose to OC. Specifically, hematinic deficiencies and a high-carbohydrate diet are said to contribute to OC development [70]. The following deficiencies were attributed to this increased risk: iron, zinc, magnesium, selenium, folic acid, and vitamins (A, B6, B12, and C) [80].

## 4. Host Immune Response

As *C. albicans* is a frequent commensal colonizer of the oral mucosa, the host immune response in the oral cavity is oriented toward a more *tolerogenic state*, to avoid an excessive inflammatory response that could be damaging to the oral tissue [103]. However, the polysaccharide-rich cell wall makes *C. albicans* highly immunogenic and easily recognized by the host pattern recognition receptors (PRRs) [18,104,105,106]. Epithelial cells, upon *Candida* recognition, induce the secretion of several antimicrobial peptides with a direct killing effect on the fungal cell, which aid in controlling local colonization [16,107,108]. Secretion of proinflammatory mediators such as cytokines and chemokines (G-CSF, GM-CSF, IL-1α, IL1β, IL-6, IL-8, and CCL5) by epithelial cells, signal the recruitment of phagocytic cells, including neutrophils, macrophages and dendritic cells (DCs) to the site of infection [109,110,111,112,113,114]. Several comprehensive reviews on *C. albicans* and host cells and the immune response during *C. albicans* mucosal infection were recently published [27,108,115]. Here, we focus on highlighting oral local innate immune defenses that play a crucial role in maintaining *Candida* in its commensal state in the oral cavity.

Oral epithelial cells are the first line of defense against *C. albicans*, functioning as a physical barrier. However, the constant flow of saliva also acts as an important mechanical clearance mechanism by preventing adherence of *Candida* to the epithelial cells and, therefore, saliva secretion is important for maintenance of the commensal state of *C. albicans* in the mouth [66,116]. Additionally, saliva is highly enriched in antimicrobial peptides (AMPs), which play a vital role in innate immunity and defense against microbial colonization [84,85,117,118,119,120]. While most AMPs are produced by several cell types, the histatins, a family of 12 histidine-rich cationic peptides with broad-spectrum antimicrobial activity, are unique in that they are exclusively produced by the salivary glands [121,122]. Among the members, histatin-5 specifically possesses potent antifungal activity, primarily against *C. albicans* [121]. The anticandidal mechanism of histatin-5 is described to involve binding to specific receptors on the fungal cell wall and intracellular uptake where it targets the mitochondria, disrupting cell homeostasis [123,124,125,126].

Given the importance of saliva, it is not surprising that salivary hypofunction is considered a predisposing factor to OC. This state of absent or diminished saliva is often a side effect of conditions that cause salivary gland dysfunction, such as head-and-neck radiation or Sjögren’s syndrome, a chronic inflammatory autoimmune disorder [86,127]. Salivary gland function was also reported to be affected in HIV^+^ and AIDS individuals who often suffer from recurrent episodes of OC; interestingly, a clinical study demonstrated significantly reduced salivary histatin-5 levels in an HIV^+^ patient population [87,128]. Another condition, hyper-IgE syndrome, a rare congenital immunodeficiency state, was also shown to cause impairment in the production of AMPs, including salivary histatins [129]. Although vastly different in etiology, individuals affected with all the aforementioned conditions are known to be highly predisposed to OC, underscoring the importance of saliva and its effectors in protection against *C. albicans* proliferation.

In terms of adaptive immunity, the importance of T cells in mediating immune response to *Candida* is clearly illustrated by the high proportion of AIDS patients with low CD4^+^ T-cell counts who develop OPC [23,130]. Naive CD4^+^ T helper (Th) cells can differentiate into three types of effector T helper cells, namely, Th1, Th2, and Th17, each secreting a different set of cytokines with specific final response outcomes [131]. Within the context of OC, a dual CD4^+^ Th1-type phagocyte-dependent response and CD4^+^ Th17-type response are the main Th subsets involved [132,133]. However, although the Th1 subset plays a pivotal role during *Candida* established and systemic infections, Th17 cells are mainly involved in mucosal host defenses, controlling initial growth of *Candida* and inhibiting tissue invasion [132,134].

The initial phases of OC are defined by a prototypical neutrophil response; Th17 cells recognize pathogen-associated molecular patterns (PAMPs) via several C-type Lectin Receptors (CLRs) and the inflammasome, releasing IL-23, IL-1β, IL-6, and TGF-β, which direct the Th17 cells to the mucosal areas. Released cytokines (IL-17A, IL-17F, IL-22) at the site of infection recruit neutrophils, amplifying the secretion of proinflammatory cytokines and chemokines [129]. Noteworthy, IL-17 and IL-22, co-expressed by Th17 cells, also cooperatively enhance the expression of AMPs such as β-defensins, calprotectin, and histatins [135,136,137].

Additionally, a subset of Th17 cells called CD4+ “natural” Th17 cells (nTh17) residing in the mucosal tissue secrete IL-1, thus mediating antifungal protective immunity in the beginning of infection, since priming of naïve Th cells into effector Th17 cells and recruitment to the site of infection take time [135,138,139]. Interestingly, proliferation of the mucosal resident nTh17 was shown to be induced by the recently identified *C. albicans* secreted toxin candidalysin, which was shown to be essential for the development of OC in mice [140]. Thus, as a component of the innate immune mucosal response, nTh17 responds to the secreted candidalysin, signaling synergistically with IL-17 via induction of IL-1 family members from epithelial cells, augmenting expression of proinflammatory cytokines [140]. The importance of the Th-17 subset of cells in the protective immunity against *C. albicans* mucosal infection is well illustrated by the excessive growth of *Candida* on the skin and mucosa of patients with chronic mucocutaneous candidiasis (CMC); these patients exhibit an autosomal recessive deficiency in the IL-17 cytokine receptor IL-17RA or an autosomal dominant deficiency of IL-17F [141]. In addition, patients with hyper-IgE syndrome (HIES) have a mutation in the transcription factor STAT3, which is important in several steps along the Th17 development pathway [142]. In all these cases, the inability to induce a Th17 response and deficient IL-17 secretion lead to insufficient recruitment of neutrophils from the bloodstream and failure in containing fungal growth on the mucosa [113].

It is important to stress the importance of a healthy oral microbiota in preventing *C. albicans* shift from a harmless commensal to an invasive pathogen, as well as the key role *C. albicans* plays in maintaining homeostasis in the oral cavity. Over the past two decades, we advanced our understanding of the complex host–*C. albicans* interactions and many of the mechanisms via which *Candida* is able to evade host immunity. Nevertheless, there remain considerable gaps in our knowledge and, therefore, in order to greatly contribute to the conception of novel therapeutic strategies, we must further enhance our understanding of our host defenses.

## 5. *Candida albicans*–Bacterial Interactions in the Oral Cavity

The oral cavity is an exceptionally complex habitat harboring unique and diverse microbial communities that co-exist in an equilibrium crucial for maintaining oral health [143,144,145,146]. Any disturbances in this ecosystem that result in dominance of one pathogenic species (dysbiosis) may lead to the development of oral disease [66,145,147,148]. Within the microbial communities, extensive inter-species interactions take place that can be synergistic, in that the presence of one organism may provide a niche for others, enhancing colonization [144,149,150]. Additionally, metabolic communications among the microbial consortia also occurs; for example, excretion of a metabolite by one organism can be used as a nutrient by other organisms [151,152]. In the oral cavity, the co-adhesion of *C. albicans* with bacteria is essential for *C. albicans* persistence and, therefore, these interactions may enhance colonization in the host [145,153,154,155].

Among the oral bacterial flora, streptococci are considered to be the primary colonizers of the oral cavity important in establishing *C. albicans* colonization. Therefore, the interaction between *C. albicans* and streptococci, largely considered synergistic in nature, is the best studied of the oral fungal–bacterial interactions. Specifically, the interactions with *Streptococcus oralis, Streptococcus mitis*, *Streptococcus gordonii*, and *Streptococcus mutans* were shown to avidly adhere to the hyphae of *C. albicans* [154,156,157,158,159,160,161,162,163]. In addition to providing adhesion sites for *C. albicans* to persist within the oral cavity, streptococci were described to provide *C. albicans* with a carbon source for growth [145]. In return, by utilizing lactic acid produced by streptococci, *C. albicans* lowers oxygen tension to levels advantageous to facultative streptococci [153,164,165,166].

Although mutually beneficial, the interactions between *C. albicans* and streptococci in the oral cavity may have repercussions to the host [161,167]. One area that gained considerable interest in recent years is the interaction between *C. albicans* and the cariogenic bacteria *S. mutans*, within the context of dental caries (or tooth decay), the most common oral disease [168,169]. Caries development is mediated by the metabolic interactions between the microbial species that make up dental plaque, the biofilm formed on the tooth surface, causing fluctuations in pH, ultimately resulting in irreversible destruction of the tooth [163,169,170,171,172]. *S. mutans* is long considered to be the main cariogenic species responsible for development of dental caries; however, there is growing evidence attributing a role for *C. albicans* in mediating dental caries development via interactions with *S. mutans* [163,165,173,174,175] (Figure 2). In fact, several studies have reported high *S. mutans* prevalence in dental biofilms where *C. albicans* resides, and more importantly, clinical studies are increasingly reporting the isolation of *C. albicans* from patients with caries [176,177,178,179]. Interestingly, high levels of sugar consumption, a common predisposing factor of dental caries, also correlates with a higher occurrence of *S. mutans–C. albicans* interactions [180].

Using a rat model of dental caries, a study by Klinke et al. [181] demonstrated that *C. albicans* is in fact capable of causing occlusal caries in rats. These observations are not surprising as, similar to *S. mutans*, *C. albicans* has the ability to produce and tolerate acids. Thus, the potential role for *C. albicans* in dental caries development, via physical and metabolic interactions with *S. mutans*, was corroborated by many lines of evidence [163,174,175,178,182,183]. Furthermore, the *S. mutans*-produced exoenzyme glucosyltransferase B (gtfB) was shown to be deposited on the surface of *C. albicans* hyphae, aiding the fungus in adhering to oral surfaces [160,184,185]. Collectively, these studies strongly indicate that the presence of *C. albicans* in the oral environment could be considered a risk factor for the development of dental caries.

*Streptococcus oralis* was also associated with *C. albicans* in vitro and in vivo; a recent study by Cavalcanti et al. [186] reported an increase in *C. albicans* filamentation and biofilm formation in the presence of *S. oralis* in vitro. Importantly, using a murine model of co-infection, Xu et al. [187] showed significantly more tongue lesions and higher proinflammatory responses in the presence of *S. oralis*. Subsequent studies indicated that this effect is partly a consequence of a synergistic activation for the increase of μ-calpain, which cleaves E-cadherin in epithelial cells, thus facilitating *C. albicans* invasion of tongue tissue [188]. Additionally, it was shown that *S. oralis* induces the expression of the *C. albicans* hyphal-specific gene *EFG1* and the Als1 adhesin, promoting co-adherence between bacterial and fungal cells [189].

*Streptococcus gordonii* is another species capable of interacting with *C. albicans* via physical interaction mediated by the binding of *S*. *gordonii* adhesin SspB to the fungal adhesin Als3p, although the bacterial adhesin CshA and the fungal adhesins Eap1p and Hwp1 were also reported to be important [33,162,190,191,192]. In addition to physical interactions, the bacterial cells were also shown to induce *C. albicans* filamentation via the secreted quorum sensing autoinducer 2 (AI-2), by inhibiting the effects of the *C. albicans* quorum sensing molecule farnesol [162]. Significantly, mixed biofilms with both species were shown to be more resistant to antimicrobial treatment compared to single-species biofilms [157,193,194]. In contrast, mixed biofilms with *C. albicans* mutant strains deficient in hyphae formation and production of exopolymeric matrix displayed lower *S. gordonii* tolerance to antibiotic treatment, suggesting that *C. albicans* may protect the bacteria in mixed biofilm [158,194].

Similar to dental caries, periodontitis is a prevalent oral disease mediated primarily by the anaerobic bacterial species *Porphyromonas gingivalis* and *Fusobacterium nucleatum* [195,196]. Although the relationship between *C. albicans* and periodontitis remains undefined, *Candida* was co-isolated from subgingival plaque from patients with periodontitis, and high *Candida* levels were shown to correlate with chronic and aggressive forms of periodontitis [197,198]. The described interactions between *C. albicans* and *P. gingivalis* are conflicting as, in one study, *P. gingivalis* was shown to induce hyphae formation producing a more invasive *C. albicans* phenotype, whereas other studies described *P. gingivalis* to exert an inhibitory effect on hyphae formation [199,200]. More recently, the *C. albicans* hyphal-specific adhesin Als3p was identified as the fungal receptor for the bacterial internalin family protein InlJ [201]. Furthermore, this fungal–bacterial interaction in anoxic conditions resulted in upregulation of the *C. albicans* secreted proteolytic enzymes Sap3 and Sap9 considered to be important virulence factors [202]. In addition to *P. gingivalis*, *Candida* was also shown to co-aggregate with the periodontal pathogen *Fusobacterium nucleatum* [203,204,205]. Interestingly, as a consequence of direct contact binding, *F. nucleatum* was shown to inhibit *C. albicans* filamentation, reducing the ability of fungal cells to escape macrophages upon phagocytosis [206]. Similarly, the presence of *C. albicans* suppressed *F. nucleatum* response to macrophage attack, suggesting that both species mutually promote commensalism [206].

The interaction between *C. albicans* and the opportunistic pathogen *Staphylococcus aureus* is another seemingly synergistic fungal–bacterial interaction that was well studied [34,207,208,209,210,211]. *S. aureus* commonly colonizes the skin and although the oral microenvironment is a transient one for staphylococci, *S. aureus* is commonly co-isolated with *C. albicans* from cases of DS and periodontitis [212,213,214,215]. Significantly, however, using a mouse model of OC, it was demonstrated that upon onset of OC, mice co-colonized with *C.*
*albicans* and *S. aureus* suffered systemic bacterial infection with high morbidity and mortality [216,217]. Similar to interactions with other bacteria, the *C. albicans* hyphal-specific adhesin Als3p was also identified to be a receptor for *S. aureus* adherence to the hyphae, mediating *S. aureus* invasion of oral mucosal barriers [217].

There is a great deal we do not understand about the influence of candidal populations on the composition of microbial communities. It is now clear, however, that *C. albicans* is not only an important component of the oral microbiota, but is also an important player in shaping the oral microbiome [161,167]. Therefore, it is crucial that we determine mechanistically the precise details of *C. albicans* adhesion and signaling under conditions of co-existence with bacterial populations and how these associations influence various aspects of host physiology and disease outcomes. To that end, future efforts should focus on clinical studies, and on designing animal model systems to study fungal–bacterial interactions in vivo, with the goal of developing novel therapeutic strategies to prevent infections through targeted actions.

## 6. Animal Models

### 6.1. Mouse Model of Oropharyngeal Candidiasis

Animal models were instrumental in our understanding of *Candida* virulence factors and the factors leading to host susceptibility to *Candida* infections; importantly, animal models provided new insights into the development of novel therapeutic approaches [76,218,219]. The demonstrated similarity to human disease processes and host immune responses made the rodent model the premier model to study *Candida* pathogenesis. In fact, a large gamut of clinically relevant animal models are available to study the various systemic or mucosal diseases caused by *Candida* [23,51,220,221,222,223].

The established model of OPC is relatively simple to use and has been well validated, making it the standard animal model to study OPC. In this model, in order to establish infection, mice are immunosuppressed by administration of subcutaneous injections of cortisone acetate prior to oral inoculation with *C. albicans* [224]. This protocol results in reproducible infection mimicking what is seen in humans, namely, pseudomembranous candidiasis (Figure 3A). Infection can be assessed clinically in mice by the presence of the white lesions typical of OC, by histopathology and microscopic analysis of infected tissue (Figure 3B,C), and quantitatively by fungal burden through culturing. Using a constitutively luciferase-expressing *C. albicans* strain, non-invasive methods were also developed for real-time monitoring of OPC progression in vivo via bioluminescence imaging [225].

A main advantage of the mouse model is that it is amenable for genetic manipulation, and development of transgenic and knockout mice with targeted immune defects can be accomplished. One example is the generation of a CD4/HIV^MutA^ transgenic mouse model that established an AIDS-like disease; this transgenic mouse model was instrumental in demonstrating that HIV-mediated loss of CD4^+^ T cells underlies the susceptibility to mucosal candidiasis [226]. The same model was also used to demonstrate that loss of IL-17- and IL-22-dependent induction of innate mucosal immunity to *C. albicans* is key to susceptibility to OPC, confirming the importance of the crosstalk between adaptive and innate mucosal immunity in maintaining *Candida* commensalism [227]. In another application, mice with conditional deletion of IL-17RA in superficial oral and esophageal epithelial cells (Il17ra^ΔK13^) were used to show that oral epithelial cells dominantly control IL-17R-dependent responses to OPC through regulation of β-defensin-3 expression [134].

Additionally, the mouse model of OPC was also adapted to study the interaction of *C. albicans* with various oral bacterial species. Using an oral *S. oralis* and *C. albicans* co-infection model, a study by Xu et al. [187] demonstrated that mucosal commensal bacteria can modify the virulence of *C. albicans* in the oral cavity, where *S. oralis* not only augmented oral lesions but also promoted deep organ dissemination of *C. albicans*. A subsequent study revealed that *S. oralis* modulates the expression of the *C. albicans ALS1* gene, promoting the fungal–bacterial interaction in the oral cavity [189]. In addition to streptococci, using the OPC mouse model, a study by Kong et al. [217] identified a novel phenomenon involving the onset of OC predisposed animals to systemic infection with *Staphylococcus aureus* with high morbidity and mortality. Importantly, the mouse model of mucosal candidiasis was recently used to establish a key role for *C. albicans* in shaping the complex resident bacterial communities and in driving mucosal dysbiosis [167].

### 6.2. Rat Model of Denture Stomatitis

In vivo models of DS are advantageous over in vitro studies, in that in vitro biofilm studies on denture materials fail to account for the presence of saliva and host immune factors. Historically, the first in vivo models of DS featured acrylic dentures on monkey palates [228]. For obvious ethical and cost issues, rodent models were employed and were found to consistently reproduce clinical DS; therefore, the rat model is established as the gold standard to study biofilm-associated *Candida* DS. Several contemporary in vivo rat models of DS exist, each with their own specific indications, advantages, and limitations [63,76,229,230,231]. Many of these models proved how closely the disease process in rats can mimic what is seen in humans. This is so because the clinical and histopathological changes of DS in several of these rat models are identical to what is seen in humans. Similar to the mouse models of OC, the rat models were also adapted and optimized for host immune response studies, gene expression studies, for studying mechanisms of *Candida* drug resistance, and for therapeutic evaluations [63,230]. However, one main disadvantage of these models is the method of retention used to secure the appliance to the rat palates, which involves tying orthodontic wires around the rat teeth [63,229,230]. Although effective in retaining the appliances in the animals, the installation procedure is time-consuming, technically challenging, and likely causes discomfort to the animals. Additionally, this method of retention does not achieve an intimate fit between the device and palate and, therefore, does not mimic denture wearing in humans. Importantly, due to the poor fit, much higher infectious doses of *C. albicans* were required to induce clinical disease. An alternate method for retaining devices is by cementing, which was shown to provide significantly longer retention rates than wire-retained devices [231].

The use of cortisone immunosuppression to establish infection varies between studies. The rationale for the administration of a single dose of cortisone on the day of infection is two-fold: (1) to lower the infectious doses needed, and (2) to ensure successful initial colonization of *C. albicans* on the devices [76]. Although immunocompetent individuals develop DS, not employing immunosuppression in the rat model necessitated the use of extremely high inocula of *C. albicans*, at cell densities that are not reflective of the normal oral environment [229,232]. One interesting model, developed by Johnson et al. [229], is unique in that the device system consists of a dual fixed and removable magnetic component, making the model amenable for longitudinal biofilm sampling.

Another common disadvantage among the existing models is that devices are custom-designed for each animal, which requires making impressions of the palate for each individual rat. Furthermore, the devices are fabricated in a dental lab and may require adjustments to ensure adequate fit, a process that is costly and time-consuming. Most recently, Sultan et al. [76] developed a three-dimensional (3D) printed digitally designed rat intraoral device with precise universal fit based on a scan taken of only one rat palate. In addition to the universal fit, a unique advantage of the 3D-printing technology is the high throughput for fabricating devices and, importantly, if needed, modifications to the design can be digitally made rapidly (Figure 4).

Several of the available rat models were adapted and used to evaluate therapeutic strategies targeting DS [63,76]. Whether involving topical or systemic administration of antifungal therapy, as expected, the models demonstrated that treatment is ineffective once mature biofilms are formed on the surface of implanted devices [63,76]. However, a recent study evaluating the efficacy of a novel antimicrobial peptide-based hydrogel formulation against OC and DS demonstrated the formulation to be efficacious in preventing disease development in a mouse model of OPC and in a rat DS model [76,219]. Combined, the findings from these studies clearly indicate the need for developing targeted preventative therapeutic strategies against biofilm-associated infections that tend to be recalcitrant to therapy, such as DS [40,233,234].

## 7. New Approaches in Antifungal Drug Discovery Anti-Virulence Drugs

Treatment of candidiasis, mucosal or invasive, relies on a limited arsenal of antifungal agents. These antifungal agents comprise three main classes: polyenes, azoles, and echinocandins [235,236]. The paucity of antifungal classes coupled with the shortcomings of the current therapeutic agents hampers our ability to fight fungal infections. The most significant shortcomings of the available agents is their suboptimal selectivity, heightened toxicity, and their increased likelihood of developing resistance. Amphotericin B specifically, although considered the “gold standard” of antifungal therapy, is inherently toxic due to its lack of selectivity, given that fungal and mammalian cells are eukaryotic and share many similar biological processes [236,237]. Azoles, such as fluconazole, lack toxicity to human cells; however, they are fungistatic drugs, and this led to the emergence of resistance [237]. The newest class of antifungals, and the first to be fungal-specific, is represented by the echinocandins (caspofungin), which target a key component in the fungal cell wall not present in mammalian cells [235,236,238]. Unfortunately, the clinical use of echinocandins is limited to the treatment of systemic candidiasis, and emergence of resistance, particularly in *C. glabrata*, is becoming a concern [235,237,239]. Therefore, there is a critical need for identifying novel drug targets to circumvent the shortcomings of the currently available antifungal agents.

One strategy geared toward accomplishing this goal is targeting specific virulence factors. In essence, an anti-virulence approach would “disarm” the pathogen, thus preventing, in the case of *C. albicans*, the transition from harmless commensal to pathogen [240]. Maintenance of a commensal state rather than eradication may in fact be a more advantageous strategy, as it lowers the propensity for the development of acquired resistance. In *C. albicans*, filamentation and biofilm formation are properties central to the pathogenesis of this opportunistic pathogen [44,52]. In fact, the majority of *Candida* infections are associated with biofilm formation, best exemplified by DS, which tends to be refractory to therapy [241]. Therefore, the prospect of antifungal drug development targeting these two key biological processes is particularly attractive, and it gained considerable attention [237]. Indeed, a large number of small molecules were shown to modulate the yeast-to-hypha conversion in *C. albicans*, such as the secreted quorum-sensing molecule farnesol, rapamycin (Tor kinase inhibitor), the Hsp90 inhibitor geldanamycin, histone deacetylase inhibitors, and cell-cycle inhibitors [242,243]. However, several of these molecules were found to impact overall growth and/or were reported to impair the cell cycle, undermining their “anti-virulence” potential.

A large-scale phenotypic screening of 20,000 small-molecule compounds from the NOVACore library identified a diazaspiro-decane structural analog compound with potent inhibitory activity on *C. albicans* filamentation and biofilm formation with no effect on overall growth. Importantly, serial exposure to this compound failed to induce resistance, and in vivo efficacy was demonstrated in a murine model of OC [244]. More recently, a large-scale phenotypic screening of 30,000 drug-like small-molecule compounds from the ChemBridge’s DIVERSet chemical library identified a novel series of bioactive compounds with a common biaryl amide core structure, able to prevent *C. albicans* filamentation and biofilm formation in vitro. The lead compound also showed promising results in preventing invasive and OC in murine models, and transcriptomic analysis confirmed the downregulation of genes associated with filamentation and virulence such as *SAP*5, *ECE*1 (candidalysin), and *ALS*3 [245,246].

Currently, a few promising compounds are in the pipeline, including two new glucan-synthesis inhibitors (SCY-078 and CD101) and an inhibitor of the fungal Gwt1, an enzyme in the glycosylphosphatidylinositol biosynthesis (GPI) pathway (APX001) [239,247,248,249]. All of these compounds showed anti-*Candida* efficacy in vitro and in vivo, and they are presently undergoing phase 1 and 2 clinical trials (ClinicalTrials.gov SCY-078: NCT0224406; CD101: NCT02734862; APX001: NCT02956499 and NCT02957929).

## 8. Concluding Remarks

In conclusion, *C. albicans* is a highly adaptable microbial species capable of causing infection at various anatomical sites. Although, over the past two decades, we advanced our understanding of the complex host–*C. albicans* interactions, there remain considerable gaps in our knowledge of *C. albicans* pathogenicity, host immune responses, and, importantly, the role of *C. albicans* as a constituent of the human microbiota. The mechanisms underlying the development of antifungal resistance continue to evolve, highlighting the critical need for developing new antifungal classes. Fortunately, significant strides in the field of antifungal discovery were achieved with promising new drugs moving to clinical trials. Moreover, advances in identifying novel bioactive compounds targeting pathogenic mechanisms, rather than growth, could prove invaluable in complementing the current antifungal arsenal.

## Figures and Tables

**Figure 1 jof-06-00015-f001:**
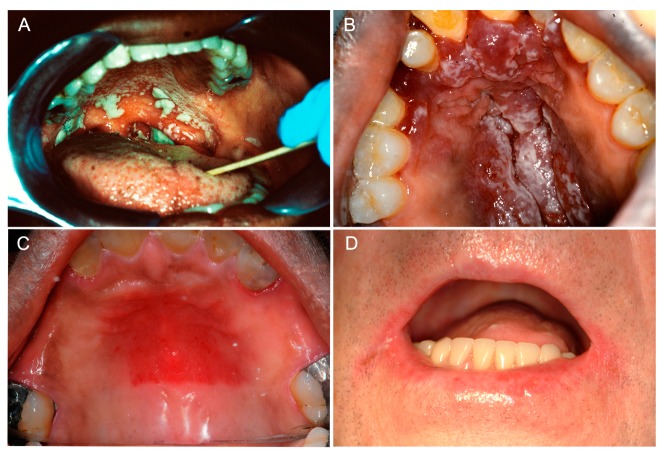
Clinical manifestations of oral candidiasis. (**A**) Oropharyngeal candidiasis characterized by diffuse thick curdy white plaques that could be wiped off with gentle scraping, with extension from the soft palatal mucosa (oral candidiasis) to the oropharynx (oropharyngeal candidiasis). (**B**) Acute pseudomembranous candidiasis in a human immunodeficiency virus (HIV)-positive individual. Multiple coalescing raised white plaques on the hard palatal mucosa on a background of underlying diffuse erythema and hyperplasia. (**C**) Newton’s class II denture stomatitis clinically manifesting as diffuse erythema of the mid hard palatal mucosa in a partial denture wearer. (**D**) Angular cheilitis presenting as bilateral erythema and maceration of the angles of the mouth. Clinical images of consented patients attending the University of Maryland School of Dentistry.

**Figure 2 jof-06-00015-f002:**
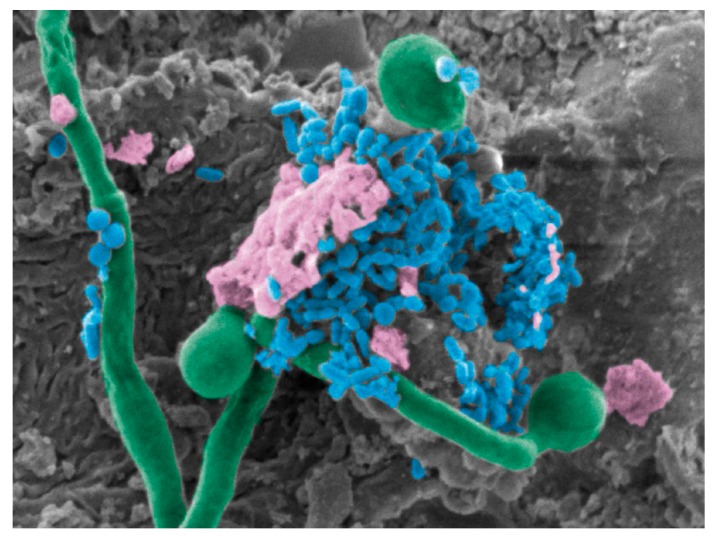
Co-infection of *Streptococcus mutans* and *Candida albicans*. SEM pseudo-colored micrograph of a mouse tongue dorsum showing *C. albicans* hyphae (green) penetrating epithelial cells with *S. mutans* (blue) cells anchoring onto hyphae within a dense extracellular matrix (pink), 10,000x magnification.

**Figure 3 jof-06-00015-f003:**
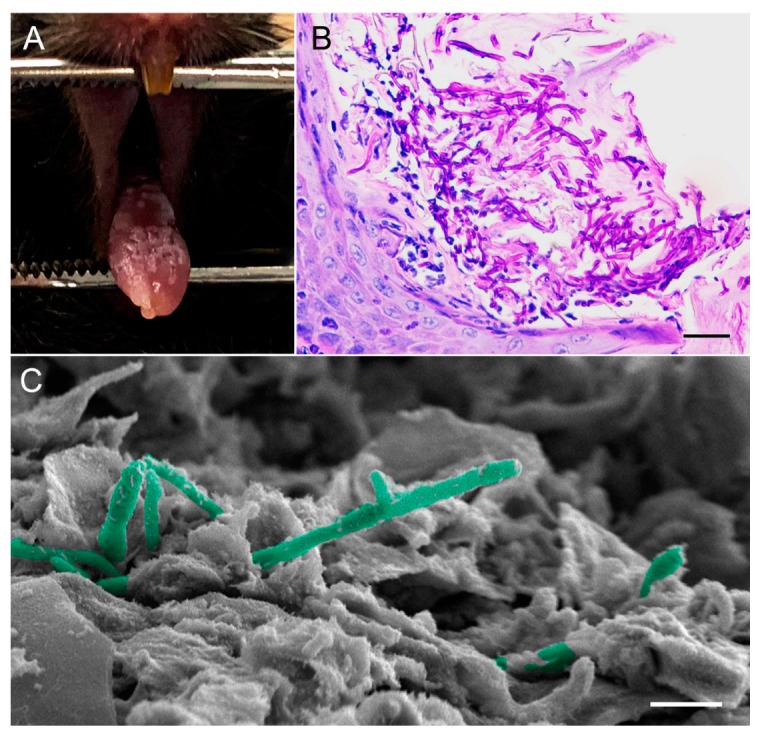
Mouse model of oral candidiasis. (**A**) Diffuse white plaques with focal areas of punched-out white lesions on the tongue dorsum of a mouse depicting oral candidiasis. (**B**) PAS-stained image showing significant accumulation and penetration of *C. albicans* hyphae through the hyperkeratotic surface of the tongue dorsal epithelium. Neutrophilic Munro microabscesses in response to the infection are seen. (**C**) SEM micrograph, high magnification of the tongue dorsum showing *C. albicans* hyphae (pseudo-colored in green) penetrating epithelial cells. Bars correspond to (**B**) 100 µm and (**C**) 20 µm.

**Figure 4 jof-06-00015-f004:**
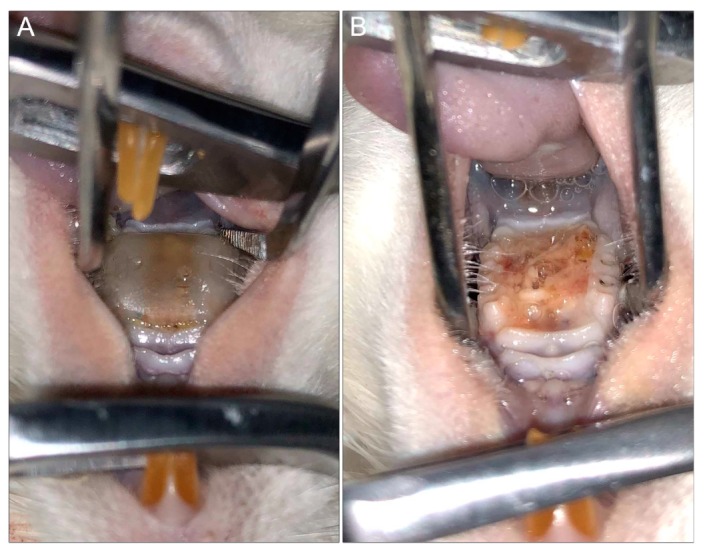
Rat model of denture stomatitis. (**A**) Three-dimensional (3D)-printed denture with intimate fit to the hard palatal-mucosa of a rat. (**B**) Denture stomatitis in a rat model demonstrating profound edema and a thick yellow biofilm of the denture-bearing hard palatal mucosa (top: tongue; bottom: hard palate).

**Table 1 jof-06-00015-t001:** *Candida albicans* pathogenic attributes relevant to oral infection.

**Adherence to Oral Epithelial Surface**
Cell surface hydrophobicity (reversible adherence) Expression of cell surface adhesins (Als3, Hwp1, etc.)
**Biofilm Formation**
Development of denture stomatitis (DS) Failure of antifungal therapy
**Evasion of Host Defenses**
Phenotypic switching Binding to complement Resistance to phagocytic stresses (oxidative and nitrosative stress response) Proteolytic degradation of host immune factors (antibodies, antimicrobial peptides, etc.)
**Invasion and Destruction of Host Tissue**
Hyphal development and thigmotropism (tissue penetration) Secretion of hydrolytic enzymes: secreted aspartyl proteinases (SAPs), phospholipases, lipases (tissue degradation) Secretion of the hypha-specific toxin candidalysin Degradation of E-cadherin Induced endocytosis

**Table 2 jof-06-00015-t002:** Predisposing factors to oral candidiasis.

**Local Factors**
Salivary dysfunction (quantitative and qualitative reductions in saliva and diminished salivary antimicrobial factors)
Poor denture hygiene and prolonged wear Ill-fitting dentures (mucosal trauma)
Topical corticosteroid therapy (steroid rinses or topical gels for management of oral mucosal disease, steroid inhalers) Smoking
**Systemic Factors**
Age-related immunosenescence (infants and elderly) Broad-spectrum antibiotics (alteration in local oral flora)
Immunosuppressive therapy (systemic corticosteroids, biologic immunomodulating agents, immunosuppressive therapies) Chemoradiation (head-and-neck cancer)
Immunocompromising conditions (thymic aplasia, hyper-immunoglobulin E (IgE)/Job’s syndrome, chronic mucocutaneous candidiasis syndromes, Sjogren’s syndrome, graft-versus-host disease, human immunodeficiency virus (HIV)/acquired immune deficiency syndrome (AIDS), leukemia)
Nutritional deficiencies (iron, zinc, magnesium, selenium, folic acid, vitamins A, B6, B12, and C) Endocrine dysfunction (diabetes, Addison’s disease, hypothyroidism)

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
