# Peer review of "Oral Candidiasis: A Disease of Opportunity"

_jof, 2020, doi:10.3390/jof6010015_

Round 1

Reviewer 1 Report

The authors submitted a well-written, comprehensive review of the current state of knowledge on oral candidiasis. The references are appropriately cited and the major works in the field have been included. The discussion of animal models is inclusive and insightful. The selected images are informative and clear. The inclusion of a brief review of novel antifungal agents under development is commendable. The manuscript does not require major revisions.

Two minor points were noted. First, the reference list has some file integrity issues that require editing (e.g. #2: Capitalize first name of author; #27, remove comma and period; #30: fix double period, etc..). Second, Figure 4 could benefit from a short sentence about the orientation to point out the animals in the photo are upside down (tongue on top, frontal incisors at the bottom).

Author Response

Response to Critiques

Rev #1

Two minor points were noted. First, the reference list has some file integrity issues that require editing (e.g. #2: Capitalize first name of author; #27, remove comma and period; #30: fix double period, etc..).

References corrected.

Second, Figure 4 could benefit from a short sentence about the orientation to point out the animals in the photo are upside down (tongue on top, frontal incisors at the bottom).

A sentence regarding orientation of image was added to the legend.

Rev #2

Overall, this is a well written and comprehensive review covering all aspects of oral Candida and oral candidiasis. My only criticism is perhaps the length – the manuscript could be shortened, particularly if the target audience of Journal Fungi, who are largely mycologists, is considered. Sections on clinical presentation, denture stomatitis and pathology which may be more suited to an oral medicine/pathology journal could be shortened. Most of this information has been reviewed and there has been little change in presentation over the last 20 years. It would be of interest whether recent oral mycobiome studies have informed us about carriage or infection with Candida species (and if not, why not?). There is little mention of non-albicans Candida. Are these irrelevant outside of the HIV population? While the new anti-fungals section is interesting, it not particularly oral relevant.  The authors have not discussed the oral relevance the existing or the prospects of resistance to current antifungals.

We do agree with the reviewer however, it was our intention to focus on the clinical aspect of oral candidiasis given the exhaustive number of reviews available on candidiasis. We felt that elaborating on the clinical aspect of candidiasis would bring new knowledge to non-clinical mycologists which could be used for future reference. We also agree with the points the reviewer mention that could be elaborated upon but as the reviewer points out, the review is already substantial in size and since these topics are well covered in other reviews, we opted to merely mention these areas of discussion and refer the readers to more detailed reviews.

 Regarding table 1 and text, what adhesins/factors are required for biofilm formation? This could be expanded. In this table we are focusing on the clinical and therapeutic implications of biofilm formation not mechanisms of biofilm formation.

 Phenotypic switching is mentioned in Table 1 but not discussed. Is there any evidence that this is really involved in immune evasion? Could be removed from the table. Yes change in morphology is described t be involved in immune evasion; this is discussed in the references provided in the text regarding morphology (line 94).

Line 191. Why italics?

Sentence corrected

Reviewer 2 Report

Overall, this is a well written and comprehensive review covering all aspects of oral Candida and oral candidiasis.

My only criticism is perhaps the length – the manuscript could be shortened, particularly if the target audience of Journal Fungi, who are largely mycologists, is considered. Sections on clinical presentation, denture stomatitis and pathology which may be more suited to an oral medicine/pathology journal could be shortened. Most of this information has been reviewed and there has been little change in presentation over the last 20 years.

It would be of interest whether recent oral mycobiome studies have informed us about carriage or infection with Candida species (and if not, why not?).

There is little mention of non-albicans Candida. Are these irrelevant outside of the HIV population?

While the new anti-fungals section is interesting, it not particularly oral relevant.  The authors have not discussed the oral relevance the existing or the prospects of resistance to current antifungals.

Regarding table 1 and text, what adhesins/factors are required for biofilm formation? This could be expanded.

Phenotypic switching is mentioned in Table 1 but not discussed. Is there any evidence that this is really involved in immune evasion? Could be removed from the table.

Line 191. Why italics?

Author Response

(The authors gave the same response as above.)
